# THE COMMON STABILITY MECHANISM BEHIND MOST SELF-SUPERVISED LEARNING APPROACHES

## ABSTRACT

Last couple of years have witnessed a tremendous progress in self-supervised learning (SSL), the success of which can be attributed to the introduction of useful inductive biases in the learning process to learn meaningful visual representations while avoiding collapse. These inductive biases and constraints manifest themselves in the form of different optimization formulations in the SSL techniques, e.g. by utilizing negative examples in a contrastive formulation, or exponential moving average and predictor in BYOL and SimSiam. In this paper, we provide a framework to explain the stability mechanism of these different SSL techniques: i) we discuss the working mechanism of contrastive techniques like SimCLR, non-contrastive techniques like BYOL, SWAV, SimSiam, Barlow Twins, and DINO; ii) we provide an argument that despite different formulations these methods implicitly optimize a similar objective function, i.e. minimizing the magnitude of the expected representation over all data samples, or the mean of the data distribution, while maximizing the magnitude of the expected representation of individual samples over different data augmentations; iii) we provide mathematical and empirical evidence to support our framework. We formulate different hypotheses and test them using the Imagenet100 dataset.

## 1 INTRODUCTION

Recent self-supervised learning (SSL) methods aim for representations invariant to strong perturbations (called augmentations) of the input image. These perturbations are changes made to an input image that are supposed to preserve the underlying semantics. Examples commonly used in SSL include random cropping, random rotation, color jittering, flipping and masking. These perturbations help the model learn to recognize the underlying structure of the image and its features, without being affected by irrelevant variations. In practice, this is done by training a projector that maps different augmentations of the same image onto the same point in the feature space, and using the gradient of the loss (the distance between the representations) to train the projector. These methods have shown to be highly effective in learning general features that can be transferred to a host of downstream tasks like classification (Van Gansbeke et al., 2020), segmentation (Van Gansbeke et al., 2021), depth-estimation (Bachmann et al., 2022), and so on.

The objective of minimizing the distance between two augmentations of the same image can lead to a trivial solution, where all images are projected onto a single point in the feature space. This phenomenon is known as *embedding collapse* (Zhang et al., 2022). Different SSL techniques use different approaches to solve this problem: contrastive SSL techniques maximize the distance between an image and other images in the dataset (called negative pairs) while minimizing the distance between the image and its augmented versions (called positive pairs). They attribute the pulling force of positive pairs to learning invariance across different augmentations, and the pushing force of negative examples to collapse avoidance (Chen et al., 2020; He et al., 2019). Non-contrastive methods do not require negative examples, and can be cluster-based, predictor-based, and redundancy minimization based. Cluster-based non-contrastive SSL (Caron et al., 2020) uses equipartitioning of cluster assignments for the collapse avoidance. Another non-contrastive SSL method SimSiam (Chen & He, 2020) uses an asymmetric student-teacher network with identical encoder architecture, and an additional projection layer, called predictor head, over the student to learn the SSL features. They claim the predictor learns the augmentation invariance. However, the exact collapse avoidance strategy of these methods is still unclear, with an empirical study pointing to a negative center vector

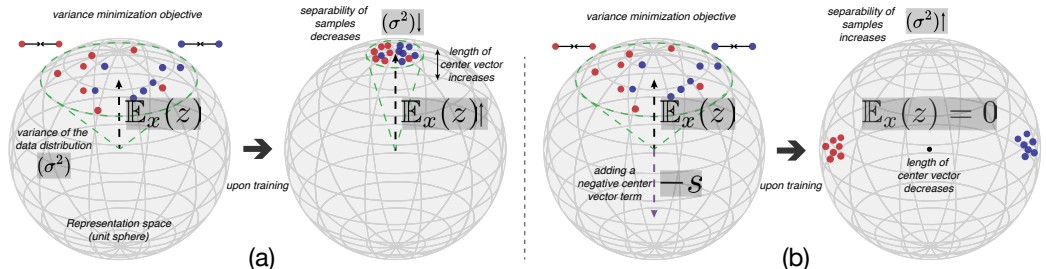

Figure 1: **Overview of our proposed learning hypothesis.** Red and blue points represent different views of two images in feature space. (a) By applying distance minimization loss between two views of the same image, the magnitude of the expected representation over the data ($\mathbb{E}_x[z]$) increases, reducing the variance of the data distribution ($\sigma^2$) in the feature space and thereby reducing their separability. (b) In order to learn a discriminative feature representation, a negative force ($-s$) equal to the expected representation over the data distribution is required. We hypothesize that this negative term is the collapse avoidance mechanism underlying different SSL methods.

gradient as a possible explanation (Zhang et al., 2022). Finally, redundancy-reduction based non-contrastive technique, Barlow Twins (Zbontar et al., 2021), uses redundancy-minimization through orthogonality constraints over the feature dimensions as a way to avoid collapse.

In this work, our goal is to uncover the underlying mechanisms behind these SSL methods. We show that they are actually instantiations of a common mathematical framework that balances training stability and augmentation invariance, as illustrated in Figure 1. We show this common framework motivates different hyperparameter and design choices that previously were set mostly empirically to obtain the best performance on downstream tasks. Our contributions are:

Major contributions:

1. We propose a single framework/meta-algorithm that explains the underlying collapse avoidance mechanism behind contrastive and non-contrastive techniques.

   - Provide a simple mathematical formulation that can explain embedding collapse for distance minimization objective (also called invariance loss).
   - Reformulating all SSL techniques showing their mathematical conversion to our proposed framework, explaining that these techniques implicitly optimize our proposed framework to avoid embedding collapse.

2. We propose a simplistic technique based on our framework, that combines distance minimization with center vector magnitude minimization as a constraint optimization problem. This also provides an empirical justification for our proposed framework.

Minor contributions:

1. We explain peculiar cases of existing SwAV with fixed prototype and Barlow twins without off-diagonal minimization in the purview of our framework.

2. We show that our proposed algorithm can be used to make predictions about, and understand rationality behind, some hyper-parameter selection in these SSL techniques which are otherwise selected purely empirically.

*Scope:* We define the scope of our work under which we explore self-supervised learning:

1. We explore contrastive and non-contrastive methods of learning representation, and do not discuss Masked-image-models (Bao et al., 2021) or other methods based on proxy task such as rotation (Gidaris et al., 2018), colorization (Zhang et al., 2016), jigsaw (Noroozi & Favaro, 2016), relative patch location (Doersch et al., 2015) etc., as they are not optimizing the feature space directly and unlikely to suffer from collapse as the ones we discuss in this paper.

2. The objective to learn the representation being invariance loss: We do not consider equivariance objectives as contemporary methods in self-supervised learning use invariance loss to minimize distance between augmented version of the input.

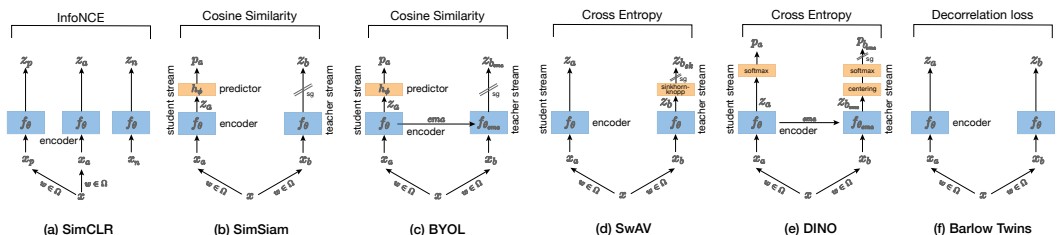

Figure 2: All methods covered by our proposed framework. For details please zoom-in.

## 2 FORMULATION

Despite the differences in the design choices and approaches, there is a unifying principle behind different SSL methods. This principle can be divided into two objectives: first, learning the augmentation invariance of the images, and second, ensuring stability in the representation space by avoiding embedding collapse. We propose that the key to stability lies in constraints imposed on the expected representation over the dataset, or what we call the *center vector*, a term coequally used by Zhang et al. (2022). These constraints prevent SSL methods from converging to trivial solutions. In particular, architectures where the optimization function minimizes the magnitude of the center vector avoid collapse, while the ones where the optimization function does not constrain it, collapse to trivial solutions. In this section, we provide a mathematical framework based on augmentation invariance and center vector constraints that generalizes different SSL approaches. We later redefine these different approaches in the purview of our framework.

### 2.1 TWO-STREAM SELF-SUPERVISED LEARNING: THE ROLE OF THE CENTER VECTOR

Let an encoder function $f$ map the RGB image space $x \in \mathcal{I}$ to the representation space $\mathbb{R}^D$ which is then normalized to the unit sphere $z \in \mathcal{S}^D$, $z = g(x) = f(x)/\|f(x)\|_2$, and $\omega : \mathcal{I} \to \mathcal{I}$ be a stochastic augmentation with distribution $P_\omega$. The center vector, $s$, is defined as the expected representation over the augmented input distribution:

$$s := \mathbb{E}_{x \sim P_x} \left[ \mathbb{E}_{\omega \sim P_\omega} \left[ g(\omega(x)) \right] \right] \tag{1}$$

For different augmentations, $\omega \in \Omega$, a two-stream self-supervised objective minimizes the distance, $\mathcal{D}$, between $z$ and the representation of the augmented version of $x$:

$$L(f) = \mathbb{E}_{x,\omega} \left[ \mathcal{D}(z, z_\omega) \right] = \mathbb{E}_{x,\omega} \left[ \mathcal{D}(g(x), g(\omega(x))) \right]. \tag{2}$$

where $z_\omega$ denotes the representation of the augmented view of the data. Here, we define our framework:

**Framework:** Optimizing only the augmentation invariance objective defined in equation 2, may lead to a trivial solution, as the representations, $z$, collapse. We posit that a non-trivial solution can be obtained by selecting an objective function that constrains the expected global representation to zero:

$$\min L(f); \quad s.t. \quad s = 0 \tag{3}$$

**Explanation:** The main objective function of distance minimization, $\mathcal{D}$, between the two views of the data is usually the Euclidean distance squared $\|z - z_\omega\|_2^2$, which is equivalent to minimizing the negative cosine similarity, $-\langle z, z_\omega \rangle$, for $L_2$ normalized vectors. Hence, the loss gradient with respect to the feature vector $z$ can be written as $\frac{\partial}{\partial z} L(f) = -\mathbb{E}_\omega[z_\omega]$. As the representation vector $z$ will move in the opposite direction of the gradient to minimize the loss, $z$ will move in the direction of $\mathbb{E}_\omega[z_\omega]$. Note that we have not considered yet the constraint $s = 0$ as $s$ must be estimated from a (stochastic) finite sample and the constrained optimization is difficult in general.

Let the expected value of $z$ over augmentations $\omega \in \Omega$, is $\mu_z = \mathbb{E}_\omega[z_\omega]$. Therefore, each term in the sum of equation 2, can be rewritten as:

$$\frac{\partial}{\partial z} L(f) = -\mathbb{E}_x[\mu_z] - (\mathbb{E}_\omega[z_\omega] - \mathbb{E}_x[\mu_z]) \tag{4}$$

The first term in the above equation leads $z$ to move in the direction of the batch center, which is common across all the samples in the batch. The second term leads it to the residual direction which is different for different samples. As $z_\omega$ is $L_2$ normalized, the magnitude of the sum of these terms is bounded above by 1, so in expectation the larger the magnitude of the expected representation, $\mathbb{E}_x[\mu_z]$ (first term) is, the smaller the magnitude of the residual representation, $\mathbb{E}_\omega[z_\omega] - \mathbb{E}_x[\mu_z]$ (second term) will be, which is not desirable. As $z$ keeps moving in the direction of the expected batch representation, its value iteratively increases and the residual vectors become smaller and smaller. All this can be avoided by minimizing the magnitude of the center vector, or the expected batch representation, $\mathbb{E}_x[\mu_z] = 0$, resulting in larger residual terms, $(\mathbb{E}_\omega[z_\omega] - \mathbb{E}_x[\mu_z])$. This residual term is different for different samples, and hence represents the semantic content of the sample. For a large batch size, the samples are representative of the global distribution of the dataset, hence the batch center $\hat{s}$ coincides with the dataset center, $\mathbb{E}_x[\mu_z]$. We define the residual component for a sample $z$ as $r_z = (z - \mathbb{E}_x[\mu_z])$.

For each iteration, explicit computation of the global center vector is non-trivial and expensive. Instead, different SSL approaches employ different ways to incorporate center vector minimization, despite being non-explicit about it. Using our framework of center vector minimization, we redefine the contemporary SSL approaches.

## 2.2 Contrastive SSL: Role of negative examples

**Triplet loss:** In contrastive SSL approaches, the goal is to bring the representations of the augmented views of an input sample close while pushing away that of other samples. To study constrastive SSL, we formulate a triplet objective function (Hoffer & Ailon, 2015). For a standard triplet loss setup, as shown in Figure 2a, $x_a$, an anchor, and $x_p = \omega(x_a)$, a positive exemplar, constitute the two views of the same data point $x_a$ and are called a positive pair, while $x_n$, a negative exemplar, is another data point and together with $x_a$ constitutes a negative pair. $z_a, z_p,$ and $z_n$ are their projections in the representation space, respectively. Then triplet loss can be written as:

$$L_{\text{triplet}}(f) = \mathbb{E}_{x_n, x_a, \omega}\left[\frac{1}{2}\max\left(\|z_a - z_p\|_2^2 - \|z_a - z_n\|_2^2 + \alpha, 0\right)\right] \tag{5}$$

In self-supervised learning, typically the two different images are pushed as far apart as possible, hence the margin $\alpha \to \inf$, which is equivalent to $\mathbb{E}_{x_n, x_a, \omega}\left[\frac{1}{2}\left(\|z_a - z_p\|_2^2 - \|z_a - z_n\|_2^2\right)\right]$ and

$$L_{\text{triplet}}(f) = \mathbb{E}_{x_n, x_a, \omega}\left[-\langle z_a, z_p\rangle + \langle z_a, z_n\rangle\right], \tag{6}$$

where the equivalence is due to the $L_2$ normalization and constants have been removed from the objective. To understand how the representation $z$ evolves, we analyze how anchors move in the representation space. To check this, we can look at the gradient of the loss w.r.t. the anchor, $z_a$. The anchor then moves in the opposite direction of this gradient.

$$\frac{\partial}{\partial z_a} L_{\text{triplet}}(f) = \mathbb{E}_{x_n, \omega}\left[-z_p + z_n\right] = \mathbb{E}_{x_n, \omega}\left[\mathbb{E}_x[\mu_z] - (z_p - \mathbb{E}_x[\mu_z]) - \mathbb{E}_x[\mu_z] + (z_n - \mathbb{E}_x[\mu_z])\right] \tag{7}$$

$$= \mathbb{E}_{x_n, \omega}\left[-s - r_p + s + r_n\right] = \mathbb{E}_{x_n, \omega}\left[-r_p + r_n\right] \tag{8}$$

$-r_p + r_n$ is the difference of the residual vectors for the positive and negative exemplars, respectively, and is desirable as it would move $z_a$ in the direction of $r_p$, the semantic component of the representation of the positive sample, and away from that of the negative sample $r_n$. If we did not have a negative sample term, $z_n$, the loss gradient would be exactly what we had in equation 4. Eventually, the center vector $s$ would become very large compared to $r_p$, as an increase in the center vector leads to a decrease in the residual as their sum is upper bounded by 1. In this case, all samples in the dataset will have high similarity to each other, since the $s$ component is present in all of them,

and the difference between any two samples, $x_i$ and $x_j$ would be small. This eventually causes collapse.

**InfoNCE:** In practice, most contrastive SSL methods use the InfoNCE Loss:

$$L_{\text{InfoNCE}}(f) = -\mathbb{E}_{x_n,x_a,\omega} \left[ \log \left( \frac{\exp(\text{sim}(z_a,z_p)/\tau)}{\exp(\text{sim}(z_a,z_p)/\tau) + \sum_{z_n} \exp(\text{sim}(z_a,z_n)/\tau)} \right) \right] \quad (9)$$

$$= -\mathbb{E}_{x_n,x_a,\omega} \left[ \text{sim}(z_a,z_p)/\tau - \text{LogSumExp}(\text{sim}(z_a,z_p)/\tau, \{\text{sim}(z_a,z_n)/\tau\}) \right]. \quad (10)$$

In SimCLR (Chen et al., 2020), a very low temperature ($\tau \ll 1$) is used. Using the identity, $\lim_{\tau \searrow 0} \tau \text{LogSumExp}(\{\text{sim}(z_a,z_i)/\tau\}) = \max_i \text{sim}(z_a,z_i)$, we see that the objective approaches

$$L_{\text{InfoNCE}}(f) \approx -\mathbb{E}_{x_n,x_a,\omega} \left[ \log \left( \frac{\exp(\text{sim}(z_a,z_p)/\tau)}{\exp(\text{sim}(z_a,z_{n_{max}})/\tau)} \right) \right] \quad (11)$$

$$= -\mathbb{E}_{x_n,x_a,\omega} \left[ \text{sim}(z_a,z_p)/\tau - \text{sim}(z_a,z_{n_{max}})/\tau \right]. \quad (12)$$

The similarity function in equation 12 most commonly used in the literature has been cosine similarity, and ignoring the constant $\tau$, the resulting objective is equivalent to:

$$L_{\text{InfoNCE}}(f) \approx -\mathbb{E}_{x_n,x_a,\omega} \left[ \langle z_a, z_p \rangle - \langle z_a, z_{n_{max}} \rangle \right]. \quad (13)$$

In summary, for normalized representation vectors, equation 13 becomes equivalent to equation 6. Hence, for InfoNCE loss as well, the stability of the representation depends on the constraint over the magnitude of the center vector. Equation 8, and the paragraph following it provide the role of positive examples for feature invariance maximization and negative examples for collapse avoidance. Further, equation 12 shows the use of temperature as a measure to sample hard-negatives.

## 2.3 SimSiam: How predictor helps avoid embedding collapse

Given the SimSiam setup in Figure 2b: $x_a = \omega_a(x)$ and $x_b = \omega_b(x)$ are two augmentations of x subjected to augmentations $\omega_a \sim P_\omega$ and $\omega_b \sim P_\omega$, respectively. $f_\theta : \mathcal{X} \to \mathcal{S}^D$ is an encoder function, parameterized by $\theta$. $h_\phi : \mathcal{S}^D \to \mathcal{S}^D$ is a predictor function, parameterized by $\phi$, such that $z_a = f_\theta(x_a)$, $z_b = f_\theta(x_b)$ and $p_a = h_\phi(z_a) = h_\phi(f_\theta(x_a))$. For any iteration $t$, these three equations and the corresponding loss are:

$$z_a^t = f_\theta^t(x_a); \quad z_b^t = f_\theta^t(x_b); \quad p_a^t = h_\phi^t(z_a) = h_\phi^t(f_\theta^t(x_a)) \quad (14)$$

$$L_{\text{SimSiam}}^t(f,h) = \frac{1}{2}\mathbb{E}_{x,\omega_a,\omega_b} \left[ -\langle p_a^t, \text{sg}(z_b^t) \rangle - \langle p_b^t, \text{sg}(z_a^t) \rangle \right] \quad (15)$$

$$= \mathbb{E}_{x,\omega_a,\omega_b} \left[ -\langle p_a^t, \text{sg}(z_b^t) \rangle \right] \quad (16)$$

where sg indicates that backpropagation will not proceed on that variable (**?**). It holds

$$2 - 2\langle p_a^t, \text{sg}(z_b^t) \rangle = \|p_a^t - \text{sg}(z_b^t)\|_2^2 = \|h_\phi^t(z_a^t) - \text{sg}(z_b^t)\|_2^2 \quad (17)$$

Now let us have a look at the predictor alone, as shown in Figure 2. Since $h_\phi^t$ has been optimized in the $t-1$ backward pass, it minimized the loss term $L_{ab}^{t-1} := \|p_a^{t-1} - \text{sg}(z_b^{t-1})\|_2^2$. The update of $h_\phi$ at the $t-1$ iteration is $\phi^t = \phi^{t-1} - \lambda \frac{\partial}{\partial \phi_{t-1}} L_{ab}^{t-1}$. SimSiam uses a high learning rate ($\lambda$) for the predictor to update it more frequently. Hence, $h_\phi$ learns to project $z_a$ to $z_b$ almost perfectly, and after the update of $h_\phi^{t-1}$ to $h_\phi^t$, we have

$$h_\phi^t(z_a^{t-1}) \approx z_b^{t-1} \quad (18)$$

After the $t-1$st update of $f_\theta$, the new updated encoder $f_\theta^t$ projects $x_a$ and $x_b$ to $z_a^t$ and $z_b^t$ respectively. This causes a shift of distribution from $P(z|\theta^{t-1},x)$ to $P(z|\theta^t,x)$, due to change in parameters from $\theta^{t-1}$ to $\theta^t$, which we denote $\mathbb{E}_{x,\omega_a}[f_\theta^t(x_a) - f_\theta^{t-1}(x_a)] = \mathbb{E}_{x,\omega_a}[\vec{\mathcal{D}}(z_a^{t-1}, z_a^t)] =: \Delta_{\text{dist}}^t$.

$$f_\theta^t(x_a) = z_a^t = z_a^{t-1} + \vec{\mathcal{D}}(z_a^{t-1}, z_a^t) \quad (19)$$

$$\Rightarrow L_{ab}^t = \|h_\phi^t(z_a^{t-1} + \vec{\mathcal{D}}(z_a^{t-1}, z_a^t)) - \text{sg}(z_b^t)\|_2^2 \quad (20)$$

Since the predictor is trained to adapt quickly to the encoder, with high learning rate (Chen & He, 2020), we assume that $h_\phi^t$ is invariant to small changes $z$:

$$h_\phi^t(z_a^{t-1} + \vec{\mathcal{D}}(z_a^{t-1}, z_a^t)) \approx h_\phi^t(z_a^{t-1}) \quad (21)$$

$$\text{expected loss becomes}: \quad \mathbb{E}_{x,\omega_a,\omega_b}[L_{ab}^t] \approx \mathbb{E}_{x,\omega_a,\omega_b}[\|h_\phi^t(z_a^{t-1}) - \mathrm{sg}(z_b^t)\|_2^2] \quad (22)$$

$$\approx \mathbb{E}_{x,\omega_b}[\|z_b^{t-1} - \mathrm{sg}(z_b^t)\|_2^2] \quad (23)$$

$$\text{where the last approximation is from eq. 18}: \quad L_{\text{SimSiam}}^t(f, h) = \mathbb{E}_{x,\omega_a,\omega_b}[L_{ab}^t] \approx \|\Delta_{\text{dist}}^t\|_2^2 \quad (24)$$

Change in the distribution between $z^{t-1}$ to $z^t$ can be written as the shift in their mean

$$\|\Delta_{\text{dist}}\|_2^2 \propto \|\mathbb{E}_{x,\omega}[z_\omega^t] - \mathbb{E}_{x,\omega}[z_\omega^{t-1}]\|_2^2 \quad (25)$$

Equation 25 suggests the expected loss at iteration $t$, causally depends on the expected representation in iteration $t-1$. We can extrapolate this to the $t = 0$, where for a randomly initialized representation space, $\mathbb{E}_{x,\omega}[z_\omega^0] \approx 0$. This causal reliance on previous iteration acts as a constraint in limiting the increase of center vector in iteration $t$.

## 2.4 BYOL: ROLE OF EXPONENTIAL MOVING AVERAGE

Similar to SimSiam, BYOL (Grill et al., 2020) is also a two-stream network with predictor on top of the online stream, Figure 2c. However, unlike SimSiam, the offline network is updated as the exponential moving average (EMA) of the online stream. Secondly, while SimSiam requires the expedite learning of the predictor with a high learning rate, BYOL uses a smaller learning rate of predictor. As a high learning rate is critical for such an architecture to avoid collapse, while BYOL still manages to avoid it, the EMA should be playing an important role in collapse avoidance.

Let the online network be $f_\theta$ and the offline network be $f_{\theta_{\text{ema}}}$, with parameter $\theta_{\text{ema}}$ updated as $\theta_{\text{ema}}^t = (1 - \epsilon)\theta^t + \epsilon\theta_{\text{ema}}^{t-1}$, with a typical value of $\epsilon = 0.99$ (Grill et al., 2020). The resulting outputs of the two views of the data points, $x_a$ and $x_b$, are $z_a$ and $z_{b_{\text{ema}}}$. Here, we can rewrite $z_{b_{ema}} = \epsilon f_{\theta_{ema}}^{t-1}(x_b) + z_{b_{ema}} - \epsilon f_{\theta_{ema}}^{t-1}(x_b) =: \epsilon f_{\theta_{ema}}^{t-1}(x_b) + z_{b_\delta}^t$.

$$L_{\text{BYOL}}(f_\theta, f_{\theta_{ema}}) = -\mathbb{E}_{x,\omega_a,\omega_b}[\langle z_a, \mathrm{sg}(z_{b_{\text{ema}}})\rangle]; \quad \frac{\partial}{\partial z_a}L_{\text{BYOL}} = -\mathbb{E}_{\omega_b}\left[\epsilon z_{b_{ema}}^{t-1} + z_{b_\delta}^t\right] \quad (26)$$

$$= -\mathbb{E}_{\omega_b}\left[\epsilon\left(r_{z_{b_{ema}}}^{t-1} + \mathbb{E}_x[\mu_{z_{b_{ema}}}^{t-1}]\right) + r_{z_{b_\delta}}^t + \mathbb{E}_x\left[\mu_{z_{b_\delta}}^t\right]\right] \quad (27)$$

$$\approx -\mathbb{E}_{\omega_b}\left[\epsilon\left(r_{z_{b_{ema}}}^{t-1} + \mathbb{E}_x[\mu_{z_{b_{ema}}}^{t-1}]\right) + r_{z_{b_\delta}}^t\right] \quad (28)$$

We see that $z_{b_\delta}^t = z_{b_{ema}} - \epsilon f_{\theta_{ema}}^{t-1}(x_b)$ will in general be close to zero, as $f_{\theta_{ema}}^{t-1}$ is close to $f_{\theta_{ema}}^t$ and $\epsilon$ is close to 1. If the initial distribution of the features are randomly distributed across the unit hypersphere, the magnitude of the center vector is 0, which the feature of each sample tries to move closer to in each subsequent iteration. This also means that no negative center vector is required in terms of the batch-level normalization etc, although they can improve the overall performance (Richemond et al., 2020). This momentum component only partially helps in minimizing the magnitude of the center vector, as discussed in the previous section, in SimSiam, the predictor helps in minimizing the magnitude of center vector as well, following a similar path of confining to the initial distribution of the feature space, that is centered to origin. It has also been shown in the literature (Richemond et al., 2020) that when the network is non-uniformly initialized leading to the non-uniform feature space, batch normalization becomes necessary to mitigate collapse. This can be explained using our derivation, that when the initial feature space is non-uniform, the center vector magnitude is non-zero, hence a negative center vector term is required to nullify the effect of $\mathbb{E}_x\left[\mu_{z_{b_{ema}}}^{t-1}\right]$.

## 2.5 DINO

DINO, a two-stream network, uses a student-teacher network similar to BYOL, with teacher parameters updated as the exponential moving average of the student, Figure 2e. Architectures of both streams are symmetric without any predictor on student network. For the output representations $z_a, z_{b_{\text{ema}}}$ corresponding to student and teacher network respectively, we can write the loss as

$$L_{\text{DINO}}(f) = -\mathbb{E}_{x,\omega_a,\omega_b}[\langle\mathrm{sg}(\tau\text{softmax}((z_{b_{\text{ema}}} - C)/\tau)), \tau\log(\text{softmax}(z_a/\tau))\rangle], \quad (29)$$

where $C$ is a momentum term based on the expected value of $z$ (Caron et al., 2021, Equation (4)). As above, taking the limit as $\tau$ goes to zero from above:

$$\lim_{\tau \searrow 0} L_{\text{DINO}}(f) = -\mathbb{E}_{x,\omega_a,\omega_b} \left[ \langle \text{sg}(e_{\max(z_{b_{\text{ema}}}-C)}), z_a - \max(z_a) \rangle \right] \tag{30}$$

$$\frac{\partial}{\partial z_a} \lim_{\tau \searrow 0} L_{\text{DINO}}(f) = -\mathbb{E}_{\omega_b} \left[ e_{\max(z_{b_{\text{ema}}}-C)} \right] + e_{\max z_a} \tag{31}$$

since $C$ is the exponential moving average of the batch centers over different iterations, this captures the notion of center vector the $s_{b_{\text{ema}}}$. Hence the above equation becomes $\frac{\partial}{\partial z_a} L_{\text{DINO}} \approx r_{b_{\text{ema}}}$. This gradient equation has a low center vector magnitude and hence the features do not move towards any certain direction and collapse is avoided. The importance of centering operation for collapse avoidance has also been studied in DINO. With this reformulation, we reexamine the centering operation in the purview of our framework to explain why it helps in collapse avoidance.

**Note:** We find that the stability of SwAV (Caron et al., 2020) and Barlow Twins (Zbontar et al., 2021), can also be explained through the center vector framework. Due to space constraint, we move the corresponding formulation sections to supplementary.

## 3 EXPERIMENTS

### 3.1 SIMPLIFIED SSL OBJECTIVE: PENALIZING CENTER VECTOR MAGNITUDE

Based on the constrained optimization problem proposed in Equation 3 we propose a simplified SSL objective:

$$L_{Simple}(f) = 0.5(L(f) - \lambda_{\mathbb{L}} s) \tag{32}$$

where $\lambda_L$ is the Lagrange multiplier, and act as a penalty term for minimizing the center vector. We optimize this unconstrained objective, $\min L_{Simple}(f)$, through mini-batch SGD. In Figure 3, we compare the performance of this simplified objective against SimSiam on toy datasets. $\lambda_L$ is a hyperparameter which we set to $-1$, however a better selection process should be possible, but is beyond the scope of this paper. We observe that our simplified objective without architectural complexity of SimSiam, is able to outperform its performance. This provides a possible justification of the proposed framework in Section 2.1.

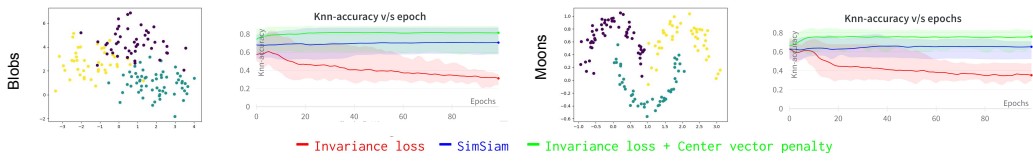

Figure 3: **Simplified SSL objective:** We show that a simplified objective that minimizes the invariance loss with a center vector penalty (green), can outperform SimSiam. We plot the toy dataset distribution on left and performance curves on right for Blob and Moons dataset. Plots are averages of five runs with varying seeds, and variance is shown by shaded regions.

### 3.2 WHY SIMSIAM COLLAPSES WITHOUT PREDICTOR? UNDERSTANDING COLLAPSE WITH TOY-DATASETS

Toy datasets provide a controlled abstraction over the complexity of the natural distribution and hence act as a test-bench for the empirical evaluation of our proposed framework. We incorporate two toy datasets: blobs, moons. Each of the datasets contains samples in 2 dimensional space for three classes, as shown in Figure 4. We treat samples of one class as the augmentations of a single image and train a SimSiam model with and without stop-gradient. Simsiam without predictor, and stop-gradient collapse for natural distribution (Chen & He, 2020) and acts as a good model to showcase the behavior of center vector for sub-optimal cases. In both SimSiam without predictor, and without stop-gradient cases, the formulation defined in equation 18, and 25 do not hold. Hence, no center vector minimization term is present in the loss, leading to collapse. Analysis on natural dataset has been provided in supplementary.

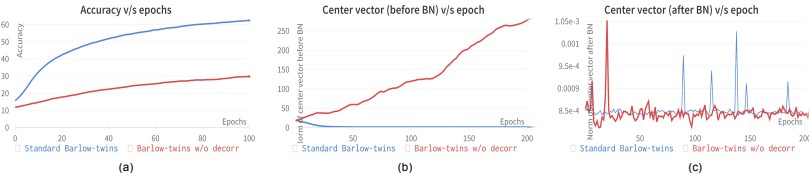

Figure 4: **Evaluation on toy datasets:** Standard SimSiam, SimSiam without predictor and SimSiam without stop-gradient have been shown in blue, red and pink respectively. Plots are averages of five runs with varying seeds, and variance is shown by shaded regions. Center vector is high for both the cases of collapse, i.e. SimSiam without predictor, and SimSiam without stop-gradient. This empirically verifies, the role of predictor and stop-gradient for collapse avoidance in SimSiam, based on our formulation. Input dataset distribution can be viewed in Figure 3

### 3.3 BARLOW-TWINS CAN WORK WITHOUT ORTHOGONALIZATION

Barlow-twins orthogonalizes cross-correlation matrices between image views. It uses an invariant loss for diagonal elements and an orthogonalization/decorrelation loss for off-diagonal elements, multiplied by a weighting factor $\lambda$. Zbontar et al. (2021) demonstrate the robustness of Barlow-twins to different $\lambda$ values. In our formulation in supplementary Section **??**, we show that pushing the off-diagonal elements to zero, is the same as minimizing the negative pair similiarity in the mini-batch. Hence, $\lambda$ should play a similar (however weaker role) as $\tau$, the temperature parameter in InfoNCE. While in InfoNCE, the $\tau$ parameter helps in sampling hard-negatives, there is no such mechanism to do so here with $\lambda$, and as per our section on Contrastive learning, hard-negative sampling is critical in minimizing the center vector magnitude and therefore in avoiding collapse. Hence, there must be some mechanism to compensate for the lack of hard-negative sampling in Barlow-twins to minimize the center vector. We argue, that Batch-normalization (BN) coupled with large batch-size in Barlow twins, helps in estimating the dataset center vector and its removal through BN. We also argue this is the reason behind the robustness against the $\lambda$ parameter in the original Barlow-twins formulation (Zbontar et al., 2021). To verify our claim, we train a Barlow-twins network without decorrelation/orthogonalization term, i.e. we only train the invariance term between two views of the data. Similar to the original implementation, we use BN. As shown in Figure 5, we see that even without decorrelation-term, or in terms of InfoNCE equivalent, without any negative pairs, Barlow-twins is able to avoid collapse, due to BN, although with suboptimal performance. This additionally empirically verifies our proposed framework of center vector minimization for SSL stability.

Figure 5: Barlow-twins can learn non-collapse features without decorrelation term in the loss formulation: (a) shows the knn accuracy of Barlow-twins with and without the decorrelation terms in the loss on Imagenet100, (b) and (c) show the norm of the center vector of $z$ before and after BN, while training Barlow-twins in the two aforementioned settings, respectively. We can see that BN helps in removing the center vector component from $z$.

### 3.4 SWAV WITH FIXED PROTOTYPES

Caron et al. (2020) show that SwAV even with fixed prototypes can learn a rich feature space, resulting in a downstream performance comparable to learnable prototypes. We analyze this fixed prototype model to investigate it in the purview of our framework. Figure 6 shows, that when the prototypes are randomly and uniformly initialized on a unit hypersphere, i.e. the $E_{z \in \text{prototypes}}[z] = 0$, the center vector magnitude is zero by design, as an inductive bias, and hence collapse is avoided. However, the manifold of the random initialized prototype space may not coincide with the natural

manifold of the data semantics, hence the performance of the learnable prototypes are better than fixed.

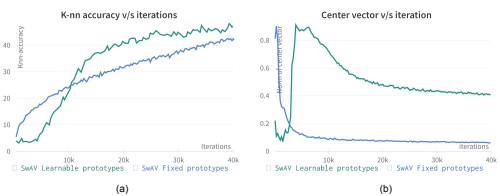

(a)  (b)

Figure 6: SwAV with fixed prototype and collapse avoidance as an inductive bias.

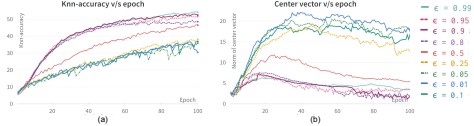

Figure 7: Analyzing the relation between different values of $\epsilon$ in EMA, on the center vector of BYOL and knn-classification on Imagenet100.

### 3.5 BYOL, IS HIGH MOMENTUM IMPORTANT?

Based on our formulation, high value of momentum $\epsilon$ is important in order to confine with the initial uniform distribution of the data in the feature space, i.e. $\mathbb{E}_x[z] \approx 0$, see Section 2.4. Here, we analyze this hypothesis, by examining the center vector and performance for different values of momentum, $\epsilon$. Figure 7 shows that lower momentum leads to higher center vector magnitudes, leading to instability and low knn-accuracies, while higher momentum leads to vice-versa, verifying the relation between center vector and stability thus performance in BYOL.

## 4 RELATED WORK

Non-contrastive methods like, SwAV (Caron et al., 2020), BYOL (Grill et al., 2020), Barlow-twins (Zbontar et al., 2021), SimSiam (Chen & He, 2020), and DINO (Caron et al., 2021) eliminate the need for negative exemplars and use a Siamese-like architecture. An online stream (student-stream) learns through gradient-based optimization, while an offline stream (teacher-stream) computes parameters based on the student's stream without direct gradient involvement.

Some of the earlier work attempting to understand the lack of collapse of non-contrastive SSL includes Tian et al. (2021), which explains the role of predictor in SimSiam as learning the eigen-space of the feature vectors. Zhang et al. (2022) introduce negative gradient of center vector as the collapse avoidance technique, however they explain it only empirically and only for SimSiam. Garrido et al. (2022) propose a dual relation between SimCLR and ViCReg (Bardes et al., 2021). While these methods provide insights about the SSL working mechanisms, they are limited to specific methods or to empirical analysis. In this work we attempt to provide a unified framework that generalizes over different contrastive and non-contrastive methods. For a detailed literature review, please refer to the supplementary.

## 5 CONCLUSION

We propose a framework for collapse avoidance in self-supervised representation learning based on center vectors. The center vector magnitude needs to be minimized to prevent feature collapse, making self-supervised feature learning an optimization problem of maximizing invariance and minimizing the expected representation. Existing self-supervised techniques can be reformulated in terms of center vector minimization. Empirical analysis on Imagenet100 and toy dataset shows that collapsed versions have higher center vector magnitudes but worse knn-classification performance compared to standard versions. We revisit SwAV with fixed-prototypes and Barlow-twins without decorrelation loss, known not to collapse, and explain their mechanisms within our framework. We propose a simplified SSL method based on our framework and our empirical evaluation supports our framework.

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
