# OpenReview forum: "The common Stability Mechanism behind most Self-Supervised Learning Approaches"
_ICLR.cc/2024/Conference — ICLR 2024 Conference Withdrawn Submission_

### Official Review · Reviewer_gdVL · 2023-10-27

**Soundness:** 3 good
**Presentation:** 2 fair
**Contribution:** 2 fair
**Rating:** 5
**Confidence:** 3

**Summary:**

The paper suggests a unifying principle that can explain stability mechanisms common to contrastive and non-contrastive SSL methods. Specifically, it identifies that the magnitude of the $\textit{center vector}$ is associated with unwanted representation collapse. The paper first shows how the objective of common SSL methods can be presented using the $\textit{center vector}$. Next, in experiments it is empirically proven that this is indeed a feature that can be associated with collapse across diverse methods.

**Strengths:**

[-] originality: In revisiting common SSL frameworks and identifying a joint underlying mechanism that allows collapse avoidance, the magnitude of the center vector, the paper presents great originality.

[-] quality: the paper provides a high-quality view of the SSL field, with detailed explanations and evaluation. It is evident that much work and effort has been put into analyzing all presented settings.

[-] clarity: the construction of the paper is such that it provides the reader with all the background, notation, and terminology, required to follow and understand the presented concepts.

[-] significance: the paper’s significance comes from uncovering the common mechanism of diverse SSL frameworks, allowing for collapse avoidance.

**Weaknesses:**

[-] contribution: while the general concept uncovered by this paper is appealing, a linkage to practical implications/applications is lacking. Vast majority of the provided experiments provide an empirical proof that for different architectures the center vector is indeed the source of collapse, however no way of utilizing this as a feature is provided. A disclaimer is the suggested “simplified SSL objective” which I relate to in the following point.

[-] “simplified SSL objective” framework: though presented as a major contribution it is studied at a very minimal level. It will be beneficial to clarify the major “simplifications” in this setting. Specifically, it will be beneficial to extend the section explaining the actual framework construction. Further, as experiments within this setting are limited to toy datasets it is unclear whether the simplifications are applicable for “real” datasets.

[-] overall structure:  summarizing the above points I believe the paper could benefit from putting more emphasis or providing more novel insights utilizing the identified role of the center vector (rather than revisiting known properties through these glasses).

[-] typos (minor): the paper could benefit from another proofread to fix sentences that seem incomplete and typos. For example, in sec. 2.3 missing math notation for $x$, citation (?), and the sentence starting with ”Now let us have a look at”.

**Questions:**

[-] novelty: to address this issue is it possible to either (i) present a more extensive evaluation of the suggested framework or (ii) highlight more properties exposed only using the center vector analysis (beyond sec. 3.3)?

[-] suggested framework: would it be possible to add a section with further elaboration on this setting and its benefits?

---

### Official Review · Reviewer_pFyt · 2023-10-28

**Soundness:** 2 fair
**Presentation:** 3 good
**Contribution:** 2 fair
**Rating:** 3
**Confidence:** 3

**Summary:**

This paper delves into the stability mechanisms underpinning various Self-Supervised Learning (SSL) methods, including SimCLR, BYOL, SimSiam, SwAV, BarlowTwins, and DINO. The authors employ a unified framework to analyze and elucidate how these methods avert collapse. Furthermore, the authors introduce a straightforward regularizer designed to penalize center vectors. To illustrate the efficacy of this approach, the paper presents results from toy experiments. Additionally, the authors engage in a discussion about the significance of fundamental components such as predictors and momentum encoders within standard SSL frameworks.

**Strengths:**

1. This paper presents a systematic study of the mechanisms at play within various frameworks. The authors also offer insightful explanations regarding the role of center vectors in preventing collapse.

2. The proposed method demonstrates simplicity and effectiveness in preliminary experiments.

3. The paper delves into several critical questions within the Self-Supervised Learning (SSL) domain, including:
(1) How SimSiam prevents collapse without a predictor.
(2) BarlowTwins without orthogonalization.
(3) SWaV with fixed prototypes.
(4) The influence of momentum in BYOL.

**Weaknesses:**

1. The level of novelty in this work appears limited. The conceptual foundation of 'preventing collapse' in Self-Supervised Learning (SSL) has already been extensively explored in the existing literature, with a more in-depth comprehension. For instance, arxiv: 2102.06810. I see limited contribution from this paper helping the community understand SSL dynamics.

2. The proposed regularizer fundamentally resembles DINO's loss, which utilizes a center vector to prevent collapse. In essence, I expect a sweeping on $\lambda$. If the proposed method really helps, one must see an U-shaped curve of perfermance vs this $\lambda$.

3. The paper lacks essential experimental details and fails to provide a compelling case for the efficacy of the proposed regularizer at the ImageNet100 level, especially without proper optimization. It would be more convincing to demonstrate its impact, possibly even on CIFAR10, with appropriate baseline results.

4. The paper appears to have been written hastily, as it contains numerous typos, missing section/figure references, and poorly explained experiment result figures.

**Questions:**

None

---

### Official Review · Reviewer_kELi · 2023-10-29

**Soundness:** 2 fair
**Presentation:** 3 good
**Contribution:** 1 poor
**Rating:** 3
**Confidence:** 5

**Summary:**

This paper is a courageous attempt at unifying understanding of most major self-supervised learning algorithms, across the two or three
families of them by now clearly identifiable. To this end, it seeks to explain non-collapse of the SSL objective by splitting it into two terms, akin to a cosine similarity term maximization with an additional term that can be seen either as a constraint or negative gradient.

**Strengths:**

The paper is fairly clearly written and easy to follow.

**Weaknesses:**

However, by now, it is a fairly well-known insight that SSL representations result from an equilibrium between the action of positive and negative terms (even if implicit), lest they collapse, and therefore that SSL losses can be split as the sum of two relevant terms [1, 2]. Hence the most critical issue in this paper is its treatment of prior work; for instance this perspective was readily known simply by generalizing contrastive methods away from log-sum-exp normalization and InfoNCE thanks to f-divergences [3] where the negative term comes from the Legendre dual representation of the objective optimized (this also generalizes easily to say SwAV). Other work have explained the negative term as coming for instance from architectural constraints [4] or from orthogonality through equivalence to gradient-based PCA [5,6,8] for non-contrastive.

Similarly the works both [7] and [8] position known SSL algorithms as instances of manifold learning and/or Riemannian SGD, which once again through its geometry explains the provenance of negative gradients. If one argues that unifying works can be assessed based on the generality of insights they provide or their empirical implications, then [7] nicely connects - or even proves equivalence of - known SSL algorithms to corresponding manifold learning algorithms; and [8] clearly explains how the stop-gradient operation is crucial in enforcing constraint in BYOL and SimSIAM, along with introducing multiple predictor-free variants based on that principle. This work's contribution unfortunately doesn't go as far in either direction as those two - and doesn't cite those published works either, thus not engaging with deep mathematical insights, or showing whether its empirical contributions are orthogonal or additive to those.

This is very relevant since observations in sections 2.2 and 2.3 have not only already been made elsewhere but also in more detail ('hϕ
learns to project za to zb almost perfectly' -> is detailed in for instance [8,9,10])... This lack of related literature research permeates all through the introduction, including the incorrect statement 'However, the exact collapse avoidance strategy of these [non-contrastive] methods is still unclear,' as it is in fact detailed and proven in [11, 8].

Two points of detail as well: first, the nested expectation minimization in the first term can also be optimized in a variety of ways, practically, and alternate maximization EM-style like in SimSIAM does not necessarily explain why a stop-gradient is necessary.
Second point of detail: the original EMA parameter in BYOL was 0.996 (with an annealing schedule all the way to 1); perhaps 'close to 1'
would be more accurate across all variants and datasets given what the authors aim to achieve. Also on the BYOL side, it is well known that weight decay is a critical factor in ensuring training stability [11,12] , so an analysis of that phenomenon rather than momentum in the optimization would be welcome.

Taken together, we feel that the contribution of this paper is not significant enough, and absolutely needs to be recontextualized using prior work. We would therefore encourage the authors to proceed to a rewrite furthering their interesting ideas, and consequently expand on the empirical results section - replicating algorithms from the below works as baseline on ImageNet-100 if compute is a concern.

[1] Wang & Isola, Understanding Contrastive Representation Learning through Alignment and Uniformity on the Hypersphere.
[2] Tao et al, Exploring the Equivalence of Siamese Self-Supervised Learning via A Unified Gradient Framework.
[3] Zhang et al, f-Mutual Information Contrastive Learning.
[4] Liu et al, Bridging the gap from asymmetry tricks to decorrelation principles in non-contrastive self-supervised learning.
[5] Tian et al, Understanding Deep Contrastive Learning via Coordinate-wise Optimization
[6] Tang et al, Understanding Self-Predictive Learning for Reinforcement Learning.
[7] Balestriero & LeCun, Contrastive and noncontrastive self-supervised learning recover global and local spectral embedding methods.
[8] Richemond et al, The Edge of Orthogonality: A Simple View of What Makes BYOL Tick.
[9] Wang et al, Towards demystifying representation learning with non-contrastive self-supervision.
[10] Zhang et al, Align Representations with Base: A New Approach to Self-Supervised Learning

[11] Tian et al, Understanding Self-Supervised Learning Dynamics without Contrastive Pairs.
[12] Grill et al, Bootstrap Your Own Latent: A new Approach to Self-Supervised Learning.

**Questions:**

A minor detail question for empirical evaluation : why no addition of top-1 and/or say top-3 (as we're on ImageNet-100) accuracy evolution, rather than just kNN-accuracy everywhere ?

---

### Official Review · Reviewer_2HcK · 2023-10-31

**Soundness:** 2 fair
**Presentation:** 2 fair
**Contribution:** 2 fair
**Rating:** 5
**Confidence:** 3

**Summary:**

The paper proposes the size or magnitude of the ``center vector'' or the mean of the dataset as being a hidden variable that is implicitly optimized (minimized) by various self-supervised learning (SSL) objectives.The paper proposes that a bias towards small center vector as an explanation of unify how several popular SSL objectives work to prevent feature collapse. Analytical explanations are provided for triplet loss, contrastive SSL (InfoNCE-based), SimSiam, BYOL, DINO and Barlow Twins. Some empirical evidence is provided with synthetic data as well as some ablations using SWAV and Barlow Twins with ImageNet-100 dataset

**Strengths:**

- The paper attempts to unify several recent SSL objectives and provide an explanation of how these methods avoid feature collapse
- The treatment starting from a simple invariance objective to triplet loss and constative SSL is simple and easy to follow
- The paper provides some empirical evidence on how the result of their analysis can be used to come up with a simplified SSL objective

**Weaknesses:**

- The paper is very light on empirical validation. The simplified SSL objective is tested on a low-dimensional problem which helps readers build intuition. However, that leaves me wondering about whether the same intuition extends to high-dimensional data
- The empirical work that uses ImageNet-100 is limited to testing certain hypotheses for a subset of methods. Suggestion here link these experiments to the analysis developed earlier in the paper would help motivate the readers about the validity and use of these experiments
- The authors may want to carefully word the scope and nature of experimental work done in the paper. If not, the reader (including me) may be expecting more empirical work than what is covered in the paper
- The analysis for SimSiam is somewhat hard to follow. One suggestion here is to include `all of the steps of the derivation in the appendix. This may aid readers of the paper to parse over results without having to struggle over some non-obvious details.
- The analysis for BYOL and DINO has similar issues.

**Questions:**

Please refer to the questions posed in **Weaknesses** section